# Parental Knowledge about Allergies and Problems with an Elimination Diet in Children Aged 3 to 6 Years

**DOI:** 10.3390/children9111693

**Published:** 2022-11-04

**Authors:** Malgorzata Kostecka, Joanna Kostecka-Jarecka, Julianna Kostecka, Katarzyna Iłowiecka, Katarzyna Kolasa, Gabriela Gutowska, Magdalena Sawic

**Affiliations:** 1Faculty of Food Science and Biotechnology, University of Life Sciences, Akademicka 15, 20-950 Lublin, Poland; 2Independent Public Healthcare Center in Łęczna, Krasnystawska 52, 21-010 Łęczna, Poland; 3Faculty of Medicine, Medical University of Lublin, Chodźki 19, 20-093 Lublin, Poland; 4Department of Food and Nutrition, Medical University of Lublin, Chodźki 4a, 20-093 Lublin, Poland

**Keywords:** allergies, allergy type, long-term elimination diet, dietary adherence, children

## Abstract

Allergic diseases are highly prevalent, and they can exert a significant influence on the patients’ physical and mental well-being, thus affecting the quality of their lives and society as a whole. The aim of this study was to evaluate parental knowledge about allergens, allergy symptoms, and treatment of allergies, and to identify problems with adherence to an elimination diet and the underlying difficulties. Twelve kindergartens and the parents of 1350 preschoolers took part in the first stage of the study. In a screening trial, allergies were diagnosed in 197 children, and their parents participated in the second stage of the study. The child’s age at the onset of the first symptoms was significantly correlated with allergy type. Age was significantly correlated with selected symptoms of an allergic reaction, and skin allergies were more prevalent in younger children. Erythema, skin reddening, and urticaria occurred more frequently in children aged 3–4 years (OR 1.45; 95%CI 1.24–1.77, *p* < 0.05) and were diagnosed in skin tests (OR 1.36; 95%CI 1.22–1.59, *p* < 0.05). Allergies to numerous food items were associated with a long-term elimination diet (OR 1.89; 95%CI 1.33–2.19, *p* < 0.01), as well as problems with preparing safe meals, shopping for food, or dietary adherence when eating out. According to the respondents, lack of support from other family members and compliance with dietary restrictions in kindergartens and when eating out posed the greatest barriers to dietary adherence. Parents do not have sufficient knowledge about environmental allergens and effective strategies for coping with acute allergic reactions, including anaphylactic shock. Children with diagnosed food allergies should enjoy a similar quality of life to their healthy peers, which is why the parents should be educated about diet therapy, duration of treatment, and safe food substitutes.

## 1. Introduction

Allergic diseases are highly prevalent, and they can exert a significant influence on the patients’ physical and mental well-being, thus affecting the quality of their lives and society as a whole. Epidemiological research [1,2] has demonstrated that the prevalence of allergies continues to increase, and new risk factors responsible for atopic/allergic diseases are being identified. The epidemiology of allergies, in particular inhaled allergies, is affected by climate and latitude. Other contributing factors include genetic predisposition, environmental pollution, allergen levels in the environment, and infections, in particular viral infections.

The following allergic reactions were identified in the 2001 position paper of the European Academy of Allergology and Clinical Immunology (EAACI): IgE-mediated (type I reactions in the Gells–Coombs classification of hypersensitivity reactions) and non-IgE-mediated (type II–IV reactions in the Gells–Coombs classification and other reactions that activate immune mechanisms) [3]. Depending on the route of allergen exposure, allergies are classified as airborne (inhaled) allergies, allergic-contact dermatitis, and food allergies.

Atopic disorders are progressive allergic conditions included in the atopic march [4,5,6]. Atopic dermatitis usually begins in infancy or early childhood, and it leads to the gradual development of food allergies, allergic rhinitis, or allergic asthma. A food allergy is a pathological response of the immune system that is triggered by the consumption of a food-protein antigen. Exposure to even very small amounts of allergenic foods can trigger clinical symptoms such as gastrointestinal disorders, urticaria, and respiratory inflammations with mild to life-threatening severity.

In Poland, the first epidemiological research on allergic diseases was undertaken in the mid-1990s under the auspices of the Polish Society of Allergology [7]. A randomized multicenter cohort study was conducted in 2006–2008 to determine the prevalence of allergies in Poland (Epidemiology of Allergic Diseases in Poland, ECAP). Three age groups were evaluated in the ECAP project: 6–7, 13–14, and 20–44 years. The questionnaire was developed based on two translated and validated screening tools: the European Community Respiratory Health Survey II (ECRHS II) and the International Study of Asthma and Allergies in Childhood (ISAAC). The ECAP study revealed that food allergies affected around 13% of children aged 6–7 years and 11% of children aged 13–14 years [8].

The aim of this study was to characterize various types of allergies in a selected group of children aged 3 to 6 years; to evaluate parental knowledge about allergens, allergy symptoms, and treatment of allergies; and to identify problems with adherence to an elimination diet and the underlying difficulties.

## 2. Materials and Methods

### 2.1. Study Design and Participants

The study was part of a research and educational project entitled “A lifelong strategy for coping with allergies—an educational program for preschoolers,” which was implemented in kindergartens in Lublin (southeastern Poland) between June 2021 and June 2022. The project involved kindergartens belonging to a local network of schools and kindergartens that promote a healthy lifestyle, as well as kindergartens that do not belong to the network but are supervised by the local department of education. Twelve kindergartens and the parents of 1350 preschoolers took part in the first stage of the study. Parents filled out a screening questionnaire regarding the presence of allergies in children. The survey involved a traditional paper questionnaire or an online questionnaire when COVID-19 restrictions were implemented in kindergartens. In a screening trial, allergies were diagnosed in 197 children, and their parents participated in the second stage of the study. The inclusion criteria were the child’s age (3–6 years) and a diagnosis of a food allergy, an inhaled allergy, or contact dermatitis. In the studied children, symptoms of allergy had disappeared with age or were still present during the study. The research tool was a questionnaire composed of 68 questions divided into three parts. The first part was based on the Polish version of the European Academy of Allergy and Clinical Immunology (EAACI) questionnaire [9] and the ISAAC questionnaire [10]. The second part (parental knowledge) and the third part (dietary habits) of the questionnaire were based on the Infant and Young Child Feeding (IYCF) assessment and the KomPAN questionnaire for evaluating dietary habits and nutrition beliefs [11,12]. All parents participated voluntarily in the study, agreed to their children’s participation in the educational program, were informed about the purpose of the study, and were assured that the study was anonymous. The questionnaire was completed independently by the parents without any assistance from the researchers.

The first part of the questionnaire contained demographic questions about the child’s gender, age, body mass, and height, as well as questions about clinical symptoms of allergy, frequency of symptoms, date of initial diagnosis, progression of allergy, diagnostic tests, results of allergy tests, specialty care and type of treatment, allergies in the family, and adherence to an elimination diet (duration, food substitutes, dietary variety, provocation tests), and an assessment of food-allergy management and eating habits. The second part of the questionnaire consisted of 15 questions on parental knowledge, and the respondents could choose one of three answers: true, false, or I don’t know. These questions assessed parental knowledge about allergens, elimination diet, cross-reactivity in allergic reactions, and food-allergen labeling. The respondents received 1 point for a correct answer and 0 points for an incorrect answer or “I don’t know” answer. Points were summed up for each respondent (range: 0 to 15 points). Based on tertile distribution, the respondents were divided into three nutrition-knowledge categories: bottom (0–7 points), middle (8–11 points), and upper tertile (12–15 points). The last part contained questions about the frequency of consumption of various food items, and the respondents could choose one of the following answers: [] never, [] 1–3 times per month, [] once a week, [] several times a week, [] once daily, [] several times a day.

### 2.2. Data Analysis

Categorical variables were presented as sample percentages (%), and continuous variables were expressed by median values and the interquartile range (IQR). The differences between groups were analyzed in the chi-squared test (categorical variables) or the Mann–Whitney test (continuous variables). Before statistical analysis, data were checked for normal distribution in the Kolmogorov–Smirnov test.

The odds ratios (ORs) and 95% confidence intervals (95% CIs) were calculated. The reference categories (OR = 1.00) included the child’s age and gender, no allergies in the family, allergy diagnosed by a specialist, and serology tests. The ORs were adjusted for the duration of the elimination diet and its safety, depending on the number of allergens. The significance of ORs was assessed with Wald’s statistics. The results of all tests were regarded as statistically significant at *p* < 0.05. Data were processed in the Statistica program (version 13.1 PL; StatSoft Inc., Tulsa, OK, USA; StatSoft, Krakow, Poland).

## 3. Results

### 3.1. General Characteristics of the Study Group

The parents of 1350 children attending 12 kindergartens participated in the study. Allergies were diagnosed in 197 children (14.59% of the studied population), including in 80 boys (40.61%). General information about the diagnosed allergies is presented in Table 1.

The child’s age at the onset of the first symptoms was significantly correlated with allergy type. Food allergies were diagnosed at the earliest age; the mean age at diagnosis was 2.3 ± 1.5 months, and gender was not a differentiating factor (*p* < 0.05). The second-earliest-diagnosed allergy was inhaled allergy, and the mean age at diagnosis was 8.6 ± 4.2 months. Contact dermatitis was diagnosed the latest, past the age of 1 year in all children, and the mean age at diagnosis was 1.2 ± 0.4 years in girls and 1.4 ± 0.2 years in boys.

The frequency of allergic reactions and its effect on pharmacological treatment were analyzed. Severe cardiovascular reactions or anaphylaxis occurred sporadically (OR 0.57; 95%CI 0.47–0.71, *p* < 0.01), whereas skin, gastrointestinal, and upper-respiratory-tract reactions were chronic conditions in most children. A total of 139 children had received pharmacological treatment, and the number of children diagnosed by a pediatrician/other pediatric specialist or an allergist was similar in this group (*p* > 0.05). However, differences were noted in the administered treatment. Pediatricians tended to prescribe antihistamine drugs for mild skin reactions and gastrointestinal symptoms, whereas allergists more frequently prescribed local and topical glucocorticoids for patients with atopic dermatitis (OR 1.87; 95%CI 1.48–2.12, *p* < 0.01). Inhaled glucocorticoids were prescribed to patients with diagnosed asthma and chronic allergic reactions (OR 1.57; 95%CI 1.34–1.96, *p* < 0.01). More than 8% of the parents in this group (16 children) had visited an emergency department at least once due to a severe allergic reaction. The incidence of emergency-department visits was significantly higher in younger children with symptoms of atopic dermatitis (*p* = 0.00012) and in older children with symptoms of anaphylaxis. The relationships between allergic symptoms, the child’s age and gender, the presence of allergies in the immediate family, diagnosis, and medical treatment were investigated to better characterize the study group. The results are presented in Table 2.

Age was significantly correlated with selected symptoms of an allergic reaction, and skin allergies were more prevalent in younger children. Erythema, skin reddening, and urticaria occurred more frequently in children aged 3–4 years (OR 1.45; 95%CI 1.24–1.77, *p* < 0.05) and were diagnosed in skin tests (OR 1.36; 95%CI 1.22–1.59, *p* < 0.05). Atopic dermatitis was also more prevalent in younger children (OR 1.74; 95%CI 1.33–1.89, *p* < 0.01), whereas gastrointestinal symptoms and anaphylaxis were more frequently reported in children aged 5–6 years. The risk of allergy was higher if immediate family members had been diagnosed with an allergic disease. The prevalence of upper-respiratory-tract reactions was significantly higher (OR 1.39; 95%CI 1.26–1.57, *p* < 0.05) in children from families with a history of allergies, whereas anaphylaxis and atopic dermatitis were much less frequently observed in this group.

### 3.2. Parental Knowledge about Allergy Management

Safe and effective management of allergies in children requires parental knowledge about the disease, the associated risks, and the presence of allergens. Parental knowledge about allergies (in points) can be influenced by various factors, including the time since diagnosis, the type of allergy, or the presence of allergies in other family members. Parental knowledge about allergy management was evaluated in the second stage of this study. The parents’ socioeconomic characteristics, including age, education, and employment status, are presented in Table 3.

The parents of children with food allergies should also be able to eliminate dietary allergens and plan safe and allergy-free diets.

The average parental knowledge score was 1.5 points higher in the group of younger children than in children aged 5–6 years (*p* < 0.05). Parents of 3–4-year-old children with cow’s-milk-protein allergy showed the best knowledge (*p* < 0.05). The parents of 3- to 4-year-olds gave a higher number of correct answers to the question regarding the safe use of non-bovine milk and plant-based milk in children suffering from cow’s-milk-protein allergy (CMPA), and the question regarding provocation tests. Parental knowledge scores were higher in the group of children with food allergies, in particular in younger children (*p* < 0.01) and in families where both parents suffered from allergies (*p* < 0.01). A third of allergic parents were classified in the upper tertile and scored at least 12 points. The parents’ education influenced their knowledge about allergies, and 85% of the parents with university education were in the upper tertile.

The ability to identify potentially allergenic foods and eliminate them from the child’s diet is an important consideration in managing food allergies. The difficulty in following the elimination diet was related not only to the parental knowledge score but also to the main allergen (Figure 1). Two-thirds of the studied parents were able to correctly identify the sources of potentially allergenic cow’s-milk proteins, more frequently younger parents (<35 years) with university education and experience of allergy in the family (*p* < 0.05). All parents were familiar with the food sources of chicken egg, fish, and citrus-fruit allergens. The identification of the sources of nuts and soy as potentially allergenic foods was most problematic. Only one-third of the surveyed parents correctly identified breakfast foods (cereals, muesli), sweets (candy bars, chocolates), and ready-made milk-based desserts as potential sources of nuts, and older parents (>35 years) with primary and secondary education were least knowledgeable in this respect (*p* < 0.05). The respondents were not aware that soy is a food emulsifier or that dietary supplements, drugs, and pharmacological products contain soy lecithin. Only 15% of the parents, mostly those with university education (*p* < 0.05), correctly identified potentially allergenic foods containing soy protein.

The surveyed parents were least familiar with the proper course of action in managing severe allergic reactions, in particularly outside the home setting and during holidays, and only a third of the parents correctly answered the question concerning the management of anaphylaxis.

### 3.3. Problems with Adherence to an Elimination Diet in Food Allergies

The parents were also assessed for their knowledge about an elimination diet. In the group of 105 children diagnosed with food allergies, only 38 children (36.2%) were allergic to one food item, a third of the children were allergic to 2–4 food items, and the remaining children were allergic to several or more than 10 food items (Figure 2).

Food allergies were most commonly triggered by cow’s-milk proteins (43.8%), chicken-egg whites (27.6%), soy (21.9%), and fruit with small seeds (20.9%), and they were least frequently triggered by celery (5.7%) and fish (5.7%). Allergies to cow’s-milk proteins and fruit with small seeds (e.g., blueberries, raspberries, black currants, gooseberries, or blackberries) were more prevalent in 3- to 4-year-olds (OR 1.57; 95%CI 1.19–1.94, *p* < 0.05), whereas allergies to fish and citrus fruit (e.g., lemons, oranges, mandarins, or grapefruit) were more frequently noted in older children (OR 1.42; 95%CI 1.20–1.67, *p* < 0.05). The prevalence of allergies to other foods was not clearly correlated with age. Allergy type and the number of allergens affected the quality of the diet and dietary adherence. The odds ratios (95% confidence interval) of the relationships between the number of identified food allergens, the duration of the elimination diet, and the safety of the elimination diet in the respondents’ opinion are presented in Table 4.

Allergies to numerous food items were associated with a long-term elimination diet (OR 1.89; 95%CI 1.33–2.19, *p* < 0.01), as well as problems with preparing safe meals, shopping for food, or dietary adherence when eating out. The parents of children who have been diagnosed with a food allergy should consult a dietician to learn about safe meal preparation and food substitutes to prevent nutrient deficiencies. Only 13 mothers had consulted a dietician within one month after the diagnosis, and mothers of children with multiple food allergies (OR 1.36; 95%CI 1.11–1.57, *p* < 0.05) and allergies to beef, poultry, and eggs (OR 1.29; 95%CI 1.09–1.50, *p* < 0.05) constituted a majority in this group of respondents. Mothers of children allergic to cow’s-milk proteins (the most prevalent food allergy) followed pediatricians’ recommendations or joined support groups on Facebook. More than 25% of the mothers had consulted a dietician at least once after diagnosis, but most of them attended only one appointment (*p* = 0.0023). Safe food substitutes ensure adequate nutrient intake. Only two-thirds of the respondents introduced food substitutes, and only one-third of the mothers whose children were allergic to two to three food items used safe substitutes (*p* = 0.0012). The parents of children allergic to one food item (mostly cow’s-milk proteins) most often used protein hydrolysates, and five mothers had completely eliminated cow’s milk and hydrolysates from their children’s diets and used plant-based milks instead.

The parents’ socioeconomic characteristics and daily behaviors can affect the safety of their children’s diets. Selected examples are presented in Table 5.

According to the respondents, lack of support from other family members and compliance with dietary restrictions in kindergartens and when eating out posed the greatest barriers to dietary adherence. The mothers of one-third of the children with food allergies reported that dietary restrictions were not observed by grandparents/cousins (*p* = 0.003) and fathers (*p* = 0.0018). Five kindergartens were reluctant to eliminate trigger foods from the child’s diet (*p* = 0.0051), and in three kindergartens, mothers were not allowed to bring safe substitutes, even when these food items were not provided by the kindergarten (*p* = 0.02).

## 4. Discussion

Childhood-onset allergies are one of the most prevalent chronic disorders in developed countries. An allergy diagnosis is always challenging for the family. Parents and other family members have to expand their knowledge about allergens and their sources, and parents of children with food allergies have to learn about allergy-causing foods and safe meal preparation. Despite the rise in the prevalence of childhood allergies, the availability of reliable knowledge about allergies has not improved, which can compromise healthy growth and development in children.

Children diagnosed with food allergies require a personalized elimination diet [13]. According to research, approximately one-third of children with food allergies are allergic to multiple food items [14,15]. In this study, children with multiple food allergies accounted for 30% of the examined population. An elimination can compromise the child’s and the family’s quality of life [16,17]; it is expensive to implement, and can be a significant source of stress when eating out [18,19,20]. An elimination diet should be introduced only when necessary. In this study, children’s nutritional status was not evaluated and anthropometric measurements were not performed, but an analysis of the results, backed by literature data, revealed that children allergic to cow’s-milk proteins and egg whites are at greater risk of protein [20] and calcium [21] deficiencies, as well as impaired growth [20,21], than healthy children and children with other allergies [20]. For this reason, children with food allergies should be monitored over time to determine whether the eliminated foods continue to trigger allergic reactions or whether they can be reintroduced to the diet. An oral provocation test is considered the gold standard in confirming a food allergy and expanding the diet in successive stages of treatment [22,23,24,25]. However, a provocation test can cause a severe allergic reaction, and the physician performing the test should have considerable clinical experience in handling acute allergic reactions. In the present study, only 50% of the children had a provocation test, and 20% of the mothers did not abandon an elimination diet after the test had confirmed that the eliminated foods no longer triggered allergic reactions. Information about clinical sensitivity is important for designing an oral provocation test and selecting the lowest possible initial dose that is safe for the child [26].

Food labels help patients adhere to an elimination diet. In the current study, 90% of the respondents were of the opinion that food labels should provide clear information about allergenic ingredients, and according to 75% of the surveyed parents, food products for allergy sufferers should be clearly labeled, for example as gluten-free or lactose-free. The food-processing industry is implementing strategies for identifying food allergens, in particular trace amounts of allergenic ingredients whose intake is difficult to estimate [27,28]. The intake of allergenic proteins should be estimated by health-care professionals based on portion size as well as the amount of allergenic proteins present in the food [28,29].

Allergic diseases significantly affect the quality of children’s life. Children with food allergies have to avoid foods containing even trace amounts of allergenic ingredients or cross-reactive allergens, which could be particularly problematic when eating out or traveling [30,31]. In the present study, more than 40% of the surveyed mothers, mostly mothers of children with food and inhaled allergies, were concerned about cross-reactivity in allergic reactions. Accidental exposure to cross-reactive allergens (pollen, house-dust mites, food allergens) can trigger severe allergic reactions [32]. Food allergies can significantly compromise children’s daily social interactions [33,34,35], and many children, in particular school-age children, experience stress and anxiety when going on holidays, participating in social events, and using public transport. Food allergies also prevent children from taking part in many family activities, which places a considerable burden on the family’s well-being and quality of life [35]. According to Rouf et al., mothers of children with food allergies develop various coping strategies for dealing with stress, practical problems, and improving the child’s and the family’s quality of life [36]. Coping strategies can generate benefits only if the parents know how to manage allergic diseases, cope with flare-ups, and avoid risk factors. Scant research on parental knowledge indicates that most parents are not fully prepared to cope with an elimination diet [37,38] or administer drugs in acute anaphylactic reactions [39]. It appears that parental knowledge affects children’s emotional well-being and shapes children’s perceptions about disease. The results of this study confirmed the observations made by Gupta et al., who reported the highest levels of parental knowledge in families that regularly visited allergists and lower levels of parental knowledge in families where children had the same symptoms but attended only regular pediatric appointments [40].

Many of the surveyed parents had experienced hostility or conflict with their relatives or other parents in their attempts to deal with their children’s food allergy. Similar observations were made in other studies, where around 25% of the respondents claimed that the child’s food allergy caused strain in their marriage and influenced the mother’s professional career [41,42]. The parents recognized the importance of the child’s food allergy and coping strategies, but many participants, most of which—similarly to the current study—were mothers, were of the opinion that they paid greater attention to the child’s health than their spouses [40]. Parents who experienced significant stress and anxiety over their children’s allergy were often pitted against their community, family, or even spouses.

Food allergies can affect a child’s performance in kindergarten and school [43]. In a US study, one-third of the respondents claimed that allergies affected their children’s school attendance, and 10% of the children in the surveyed group were home schooled due to food allergy [31]. Two studies conducted in the Netherlands demonstrated that children with food allergies missed more school days than healthy students [44,45]. In the present study, the parents reported that half of the kindergartens did not offer safe-substitute meals, and 20% of the children missed kindergarten, were picked up earlier, or were taken to kindergarten at a later hour so that they could eat a safe meal at home. Education, awareness raising, and training for kindergarten personnel appear to be essential to minimize parental anxiety and decrease absenteeism.

### Strengths and Limitations

The strength of this study is the large sample size that was representative of the population of parents of allergic preschoolers, which supported a reliable determination of the prevalence of various allergies in kindergarten children. Validated questionnaires were applied to analyze the relationships between different allergies and symptoms, diagnostic methods, treatments, and dietary strategies for allergy management, which is a novel approach in research on Polish preschoolers. One of the limitations of the study was that the prevalence of various types of allergies could not be accurately determined due to the COVID-19 pandemic, which limited access to family doctors and specialists, including allergists. In many cases (around 7% of the parents surveyed in the first stage), the parents reported various symptoms that could be associated with allergies. However, a correct diagnosis was problematic due to long wait times for a medical appointment (up to 12 months) or limited contact with a specialist (teleconsultation).

## 5. Conclusions

Most parents are not prepared to manage their children’s allergies.

Parents do not have sufficient knowledge about environmental allergens and effective strategies for coping with acute allergic reactions, including anaphylactic shock.

Allergic parents and parents of younger children with a wider range of allergy symptoms received higher knowledge scores.

The parents of children diagnosed with food allergies and sensitivity to cross-reactive food allergens should be obliged to visit a dietician. The support offered by social-media groups is important, but it cannot replace a consultation with a specialist.

Children with diagnosed food allergies should enjoy a similar quality of life to their healthy peers, which is why the parents should be educated about diet therapy, duration of treatment, and safe food substitutes.

## Figures and Tables

**Figure 1 children-09-01693-f001:**
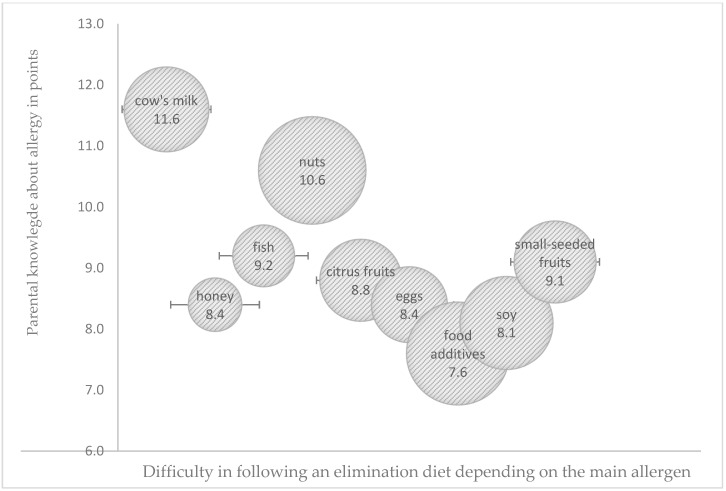
Influence of parental knowledge score (in points) on difficulties in following an elimination diet depending on the main allergen (the size of the bubble corresponds to the degree of difficulty in following the elimination diet as perceived by parents).

**Figure 2 children-09-01693-f002:**
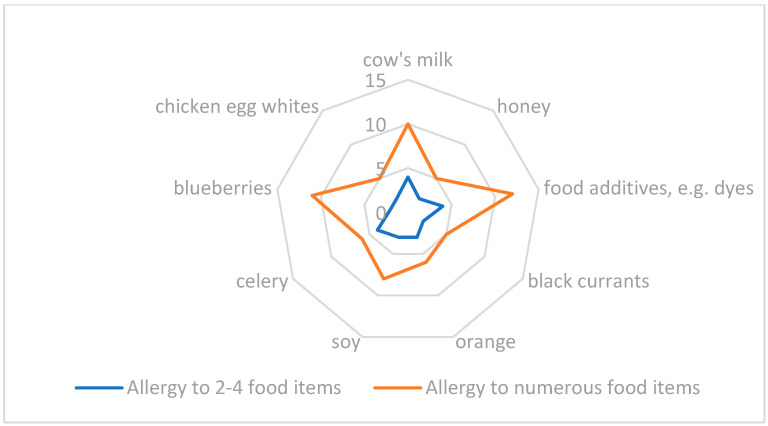
Main allergen and allergy to many food items.

**Table 1 children-09-01693-t001:** Demographic questions concerning children with diagnosed allergies.

Parameter	Total	Gender		*p*
		Male n [%]	Female n [%]	
Age				
3–4 years	93	44 [47.31%]	49 [52.69%]	<0.05
5–6 years	104	36 [34.61%]	68 [65.39%]	<0.05
Age at diagnosis (in months)	9.4 ± 5.7	9.2 ± 5.5	9.7 ± 5.9	>0.05
Type of allergy *				
Food allergy, including:	105	43 [40.9%]	62 [59.1%]	<0.05
Cow’s-milk protein	46	21 [45.6%]	25 [54.4%]	<0.05
Chicken-egg whites	29	10 [34.5%]	19 [65.5%]	<0.05
Soy	23	11 [47.8%]	12 [52.2%]	>0.05
Small-seeded fruits	22	13 [59.1%]	9 [40.9%]	<0.05
Citrus fruits	20	14 [70%]	6 [30%]	<0.05
Nuts	17	5 [29.4%]	12 [70.6%]	<0.05
Cacao	15	7 [46.7%]	8 [53.3%]	>0.05
Honey	11	3 [27.3%]	8 [72.7%]	<0.05
Food dyes	10	7 [70%]	3 [30%]	<0.05
Celery	6	4 [66.7%]	2 [33.3%]	<0.05
Fish	6	3 [50%]	3 [50%]	>0.05
Inhaled (airborne) allergy, including:	145	56 [38.8%]	89 [61.2%]	<0.005
House-dust mites	91	44 [48.4%]	47 [51.6%]	>0.05
Birch pollen	84	37 [44%]	47 [56%]	<0.05
Hazel pollen	72	51 [70.8%]	21 [29.2%]	<0.005
Grass pollen	70	37 [52.9%]	33 [47.1%]	>0.05
Mugwort pollen	34	17 [50%]	17 [50%]	>0.05
Other	56	24 [42.9%]	32 [57.1%]	<0.05
Contact dermatitis, including:	58	21 [36.2%]	37 [63.8%]	<0.005
Dog or cat hair	43	24 [55.8%]	19 [44.2%]	<0.05
Hair of other animals	27	11 [40.7%]	16 [59.3%]	<0.05
Ears of cereals	11	3 [27.3%]	8 [72.7%]	<0.005
Allergies in the family				
None	57	19 [33.3%]	38 [66.7%]	<0.005
One parent	54	27 [50%]	27 [50%]	>0.05
Both parents	47	25 [53.2%]	22 [46.8%]	>0.05
Other family members	39	9 [23.1%]	30 [76.9%]	<0.005

* Multiple answers possible.

**Table 2 children-09-01693-t002:** Odds ratios (95% confidence interval) of the relationships between selected allergy-related factors and symptoms.

	Child’s Age (Ref. 5–6 Years) 3–4 Years	Child’s Gender (Ref. Girls) Boys	Allergies in the Family (Ref. No Allergies in the Family)	Allergy Diagnosed by a Pediatrician or Other Specialist (Ref. Allergist)	Allergy Skin Test (Ref. Serology Test)	Applied Treatment—Steroids (Ref. Antihistamines)
Contact dermatitis: skin redness, erythema, pruritus, urticaria, eczema	1.45 * (1.24–1.77)	1.12 (1.03–1.29)	1.04 (0.96–1.12)	0.84 * (0.72–1.05)	1.36 * (1.22–1.59)	0.71 * (0.62–0.89)
Contact dermatitis: atopic dermatitis	1.74 ** (1.33–1.89)	1.07 (0.88–1.23)	0.78 * (0.64–0.93)	1.37 * (1.19–1.51)	0.71 * (0.62–0.89)	1.87 ** (1.42–2.06)
Oral allergy: itchiness or swelling of lips/tongue/throat, laryngeal edema, globus sensation	1.07 (0.96–1.27)	0.94 (0.88–1.07)	1.11 (1.02–1.26)	1.10 (1.01–1.24)	0.75 * (0.67–0.91)	0.63 ** (0.51–0.83)
Gastrointestinal allergy: abdominal pain, nausea, vomiting, diarrhea, gas, gastroesophageal reflux	0.74 * (0.62–1.02)	1.29 * (1.17–1.40)	1.05 (0.99–1.17)	1.14 (0.97–1.20)	1.08 (0.94–1.20)	0.47 ** (0.32–0.70)
Upper-respiratory-tract reactions: nasal pruritus, sneezing, nasal/postnasal drip, cough, hoarseness, tightness in the chest, wheezing, dyspnea	1.11 (1.02–1.24)	1.06 (0.94–1.18)	1.39 * (1.26–1.57)	1.84 ** (1.29–2.09)	1.76 ** (1.54–1.89)	1.57 ** (1.24–1.86)
Cardiovascular reactions: dizziness, hypotension, loss of consciousness	1.03 (0.93–1.11)	0.76 * (0.64–0.89)	1.03 (0.92–1.14)	1.77 ** (1.23–1.89)	1.12 (0.96–1.34)	1.05 (1.01–1.08)
Anaphylaxis	0.72 * (0.65–0.91)	1.18 (1.06–1.32)	0.54 ** (0.41–0.76)	1.34 * (1.11–1.45)	0.71 * (058–0.88)	1.02 (0.95–1.08)
Other: skin pallor, diaphoresis, fatigue, retarded growth, etc.	1.04 (0.89–1.10)	0.95 (0.82–1.11)	1.03 (0.94–1.09)	1.07 (0.91–1.15)	1.09 (1.01–1.24)	1.04 (0.91–1.12)

The odds ratios were adjusted for the allergy-related factors and symptoms, excluding the modeled variable from the confounder set. Three variables were identified as potential confounding factors: age, diagnosis, and the treatment recommended by a physician. The selection of moderators was evidence based: Age is a key variable influencing the time of symptom onset. The participants’ age was determined with an accuracy of six months and rounded to the nearest full year. The diagnosis made by an allergist is based on testing and research, and it is more reliable than a diagnosis made by a pediatrician or other pediatric specialist who was classified as another specialist. All drugs containing any type of steroids were classified as steroid treatment, whereas chromones and all drugs inhibiting histamine release were classified as antihistamine treatment. * *p* < 0.05, ** *p* < 0.01—level of significance assessed by Wald’s test.

**Table 3 children-09-01693-t003:** Allergy-knowledge score (in points) and factors that influence parental knowledge about allergies.

	Total Sample	CHILD’S AGE		
		Parents of 3–4-Year-Olds	Parents of 5–6-Year-Olds	*p*
Mean score	9.8	10.6	9.1	*
Upper tertile	49	27	22	*
Middle tertile	81	48	33	*
Bottom tertile	67	18	49	**
Allergy type				
Food	10.5	11.2	9.6	**
Cow’s-milk protein	10.9	11.7	9.3	**
Chicken-egg whites	9.4	9.2	9.7	
Inhaled	9.7	9.2	9.9	
House-dust mites	9.5	9.3	9.7	
Pollen of trees	10.8	9.2	11.0	**
Contact dermatitis	8.4	8.6	8.0	*
Dog or cat hair	8.5	8.3	8.6	
Allergies in the family				
None	7.8	7.6	8.1	
One parent	9.6	9.4	9.7	
Both parents	11.8	12.4	11.1	**
Other family members	9.7	9.1	10.6	**
Children’s age at diagnosis (in months)				
Under 6 months	11.3	11.9	10.7	*
Over 6 months	9.7	10.4	9.5	*
Parental age				
<35 years	9.7	10.2	9.1	*
>35 years	9.9	10.5	9.4	*
Parental education				
Primary/vocational	7.6	7.4	7.6	
Secondary	8.5	8.7	8.3	
University	11.6	12.7	11.1	**
Employment				
Unemployed/on prolonged leave	9.1	9.3	9.0	
Employed	10.5	11.2	10.3	*

Bottom tertile (0–7 points), middle tertile (8–11 points), upper tertile (12–15 points); *p* < 0.05—*; *p* < 0.01 **.

**Table 4 children-09-01693-t004:** Odds ratios (95% confidence interval) of the relationships between the number of identified food allergens and selected lifestyle behaviors.

	Allergy to 1 Food Item	Allergy to 2–4 Food Items	Allergy to Numerous Food Items
Allergenic foods were eliminated from the diet for more than 6 months (ref. for 3 months or less).	1.08 (0.84–1.19)	1.39 * (1.20–1.54)	1.89 ** (1.33–2.19)
The elimination diet was recommended by a pediatrician and consulted with a dietician (ref. the elimination diet was recommended by a pediatrician, but not consulted with a dietician).	1.02 (0.89–1.16)	1.13 (1.02–1.24)	1.36 * (1.11–1.57)
Allergenic foods were replaced with safe substitutes to minimize the loss of nutrients (ref. safe substitutes were not used).	1.09 (1.01–1.23)	0.81 * (0.67–0.96)	1.54 * (1.39–1.78)
When shopping for food, I always read food allergen labels (ref. I sporadically read food allergen labels).	1.13 (0.97–1.24)	1.36 * (1.09–1.57)	1.79 ** (1.41–2.27)
I always buy a different product if a given food product contains allergenic ingredients (ref. I sporadically buy other food products).	074 * (0.63–0.91)	1.05 (0.93–1.27)	1.38 * (1.22–1.59)
I often buy food products that may contain trace amounts of allergenic ingredients (ref. I never buy such products).	1.04 (0.93–1.15)	1.30 * (1.19–1.43)	0.51 ** (0.41–0.72)
Frequent consumption of small amounts of allergenic foods does not pose a health threat (ref. both sporadic and frequent consumption of allergenic foods is dangerous for the child).	1.37 * (1.15–1.49)	1.08 (1.01–1.14)	1.14 (1.03–1.26)

The odds ratios were adjusted for the duration of the elimination diet and its safety, depending on the number of allergens, excluding the modeled variable from the confounder set. The confounding factor was the classification and correct assessment of the number of allergens in the child’s diet. For example, one allergen was defined as a food product containing cow’s-milk proteins, egg proteins, soybeans, nuts, fish, and wheat allergens, as well as other products containing allergens other than those mentioned. * *p* < 0.05, ** *p* < 0.01—level of significance assessed by Wald’s test.

**Table 5 children-09-01693-t005:** Odds ratios (95% confidence interval) of the relationships between parental characteristics, the safety of their children’s diets, and problems with adhering to an elimination diet.

	Parental Education—Primary Education (Ref. University Education)	Parental Age—>35 Years (Ref. <35 Years)	Allergies in the Family (Ref. No Allergies in the Immediate Family)	Parental Knowledge—Parents in the Bottom Tertile (Ref. Parents in the Upper Tertile)
The elimination diet was recommended by a pediatrician and consulted with a dietician.	0.59 ** (0.51–0.68)	1.07 (0.96–1.11)	1.03 (0.94–1.14)	0.74 * (0.69–0.93)
Allergenic foods are replaced with safe substitutes to minimize the loss of nutrients.	1.34 * (1.27–1.56)	1.09 (1.01–1.17)	0.91 (0.82–1.04)	1.49 * (1.23–1.65)
When shopping for food, I always read food allergen labels.	0.77 * (0.54–0.91)	1.39 * (1.21–1.56)	0.96 (0.84–1.07)	1.31 * (1.25–1.48)
I sporadically buy other food products than those with listed ingredients.	1.21 * (1.03–1.37)	1.11 (0.94–1.27)	0.82 * (0.73–0.97)	1.13 (0.94–1.21)
I often buy food products that may contain trace amounts of allergenic ingredients.	1.11 (0.95–1.28)	1.27 * (1.12–1.42)	1.04 (0.92–1.23)	0.74 * (0.61–0.93)
Foods consumed outside the home setting are not safe because they often contain allergens.	1.54 * (1.27–1.87)	1.14 (0.96–1.27)	0.92 (0.76–1.09)	1.76 ** (1.54–1.89)
Both sporadic and frequent consumption of allergens is dangerous for the child.	0.88 (0.69–1.04)	0.76 * (0.61–0.95)	1.09 (0.91–1.25)	0.59 ** (0.41–0.78)

The odds ratios were adjusted for the safety of the elimination diet, depending on the parents’ socioeconomic characteristics, excluding the modeled variable from the confounder set. Two variables were identified as potential confounding factors: parental age and education. Parental age was rounded to the nearest full year, below or above 35 years. University education was defined as the acquisition of a bachelor’s/engineer’s or a master’s degree. Parents with primary education were defined as those who did not pass a matriculation exam. *p* < 0.05—*; *p* < 0.01 **—level of significance assessed by Wald’s test.

## Data Availability

Due to ethical restrictions and participant confidentiality, data cannot be made publicly available.

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
