# Peer review of "Parental Knowledge about Allergies and Problems with an Elimination Diet in Children Aged 3 to 6 Years"

_children, 2022, doi:10.3390/children9111693_

Round 1
Reviewer 1 Report
This article outlines the Polish parental experience about allergies and the problems that can occur in administering an elimination diet in children aged 3-6 years.
The study characterises various types of allergies in a selected group of children aged 3 to 6 years. It evaluates parental knowledge about allergens, allergy symptoms, and treatment of allergies, and identifies problems with adherence to an elimination diet and the underlying difficulties.
The article is clear, relevant, and presented in a well-structured manner. It is scientifically sound, and the experimental design is appropriate to investigate the parental knowledge of children with food allergies. References are relevant to the subject matter.
The article covers the range areas of an allergy diagnosis that are challenging for families.
There are just a few areas in the article where the results could be expanded to assist the reader in the clinical application of the findings.
In the results section, an overview is given in Table 1 of the demographic questions concerning children with diagnosed allergies. It would be instructive to list in detail the percentages of all types of food allergies and inhaled (airborne) allergies in this population as well as their overall total percentage.
With the text in section 3.2 regarding parental knowledge about allergy management and Table 3 it would be informative to list all the types of food allergies and inhaled allergies in detail not just the overall total percentage.
Regarding the use of the expression “allergizing” foods in the paragraph below it could more correctly be expressed as “allergenic” foods.
“The ability to identify potentially allergizing foods and eliminate them from the child’s diet is an important consideration in managing food allergies. Two-thirds of the studied parents were able to correctly identify the sources of potentially allergizing cow’s milk proteins, more frequently younger parents (< 35 years) with university education and experience of allergy in the family (p<0.05). All parents were familiar with the food sources of chicken egg, fish, and citrus fruit allergens. The identification of the sources of nuts and soy as potentially allergizing foods was most problematic.”
In section 3.2. Problems with adherence to an elimination diet in food allergies, this should be itemised as 3.3 Problems with adherence to an elimination diet in food allergies as it is a separate subsection section.
In the article it would be instructive to know what types of food allergens were implicated and what is meant by “fruit with small seeds” or “citrus fruit”. This is important as the authors point out that “Allergy type and the number of allergens affected the quality of the diet and dietary adherence”.
The paper reports that most parents are not prepared to manage their children’s allergies and the conclusion at the end of the article is comprehensive and useful for health practitioners in their clinical practice.
Author Response
We would like to thank Reviewer 1 for her/his valuable comments and efforts towards improving our manuscript. In the revised manuscript, the comments and suggestions from Reviewer 1 and our efforts to address these concerns were made in red font.
|
|
|
|
|
In the results section, an overview is given in Table 1 of the demographic questions concerning children with diagnosed allergies. It would be instructive to list in detail the percentages of all types of food allergies and inhaled (airborne) allergies in this population as well as their overall total percentage |
thank you very much for the suggestion, the table has been completed |
Table 1 |
|
With the text in section 3.2 regarding parental knowledge about allergy management and Table 3 it would be informative to list all the types of food allergies and inhaled allergies in detail not just the overall total percentage |
thank you very much for the suggestion, the table has been completed |
Table 3
|
|
Regarding the use of the expression “allergizing” foods in the paragraph below it could more correctly be expressed as “allergenic” foods The ability to identify potentially allergizing foods and eliminate them from the child’s diet is an important consideration in managing food allergies. Two-thirds of the studied parents were able to correctly identify the sources of potentially allergizing cow’s milk proteins, more frequently younger parents (< 35 years) with university education and experience of allergy in the family (p<0.05). All parents were familiar with the food sources of chicken egg, fish, and citrus fruit allergens. The identification of the sources of nuts and soy as potentially allergizing foods was most problematic |
Thank you for this right remark, we have corrected the phrase Allergenic foods in the text |
|
|
In section 3.2. Problems with adherence to an elimination diet in food allergies, this should be itemised as 3.3 Problems with adherence to an elimination diet in food allergies as it is a separate subsection section.
|
Thank you for that right remark, we corrected the subsection number, it was an editorial mistake |
|
|
In the article it would be instructive to know what types of food allergens were implicated and what is meant by “fruit with small seeds” or “citrus fruit”. This is important as the authors point out that “Allergy type and the number of allergens affected the quality of the diet and dietary adherence” |
Thank you for this important attention, we have given examples of small-seeded fruits and citrus fruits, which were the most common allergenic factors in the study group |
|
|
The paper reports that most parents are not prepared to manage their children’s allergies and the conclusion at the end of the article is comprehensive and useful for health practitioners in their clinical practice
|
Thank you very much for all your valuable comments that allowed us to improve our work. We hope that the obtained conclusions will be able to help nutritionists and other health care professionals to work better with patients. |
|
Reviewer 2 Report
The review entitled “Parental knowledge about allergies and problems with an elimination diet in children aged 3 to 6 years” by Malgorzata Kostecka and coworkers is a good compilation of data to evaluate parental knowledge about allergens, allergy symptoms, and treatment of allergies in a polish population.
The length of the manuscript, introduction, and references are adequate. The sections on data presentation are generally clear, although the reading remains monotonous in some parts. Therefore even if the the quality of the information shown makes this manuscript acceptable for publication, we suggest to change some tables into figures to make the manuscript more attractive.
We have detected editing errors as extra spaces between paragraphs (line 21-22) that will be easily corrected.
Author Response
We would like to thank Reviewer 2 for her/his valuable comments and efforts towards improving our manuscript. In the revised manuscript, the comments and suggestions from Reviewer 2 and our efforts to address these concerns were made in green font
|
The review entitled “Parental knowledge about allergies and problems with an elimination diet in children aged 3 to 6 years” by Malgorzata Kostecka and coworkers is a good compilation of data to evaluate parental knowledge about allergens, allergy symptoms, and treatment of allergies in a polish population. |
Thank you very much for the positive feedback on our work. It is very important to us because we tried to make the work at a good substantive level and interesting for the recipient. |
|
|
The length of the manuscript, introduction, and references are adequate. The sections on data presentation are generally clear, although the reading remains monotonous in some parts. Therefore even if the the quality of the information shown makes this manuscript acceptable for publication, we suggest to change some tables into figures to make the manuscript more attractive. |
Thank you for this valuable attention. We added two figures and, at the request of the second reviewer, we extended the data in the tables. |
Figure 1 and 2 |
|
We have detected editing errors as extra spaces between paragraphs (line 21-22) that will be easily corrected.
|
extra spaces between paragraphs (lines 21-22) have been corrected. |
|